# PACE: PHYSICS INFORMED UNCERTAINTY AWARE CLIMATE EMULATOR

## ABSTRACT

Climate models serve as critical tools for evaluating the effects of climate change and projecting future climate scenarios. However, the reliance on numerical simulations of physical equations renders them computationally intensive and inefficient. While deep learning methodologies have made significant progress in weather forecasting, they are still unstable for climate emulation tasks. Here, we propose **PACE**, a lightweight 684K parameter **P**hysics Informed Uncertainty **A**ware **C**limate **E**mulator. PACE emulates temperature and precipitation stably for 86 years while only being trained on greenhouse gas emissions data. We incorporate a fundamental physical law of advection-diffusion in PACE accounting for boundary conditions and empirically estimating the diffusion co-efficient and flow velocities from emissions data. PACE has been trained on 15 climate models provided by ClimateSet outperforming baselines across most of the climate models and advancing a new state of the art in a climate diagnostic task. Our code is available at `https://anonymous.4open.science/r/PACE-6874/`

## 1 INTRODUCTION

The past decade has seen superior performing data-driven weather forecasting models Kochkov et al. (2024); Lam et al. (2023); Nguyen et al. (2023b) as compared to numerical weather prediction models (ECMWF, 2023). However, the medium range forecasting ability makes them unstable for climate modelling several years into the future (Chattopadhyay & Hassanzadeh, 2023).

Climate models are governed by temporal partial differential equations (PDEs) to describe complex physical processes Gupta & Brandstetter (2022), enabling simulations of climate behavior under various forcing scenarios, such as fluctuating greenhouse gas (GHG) emissions. The computational expense associated with solving these PDEs involves, executing these climate model simulations typically for several months (Balaji et al., 2017).

In order to faithfully emulate the reference climate model, a Machine Learning (ML) based climate emulator should follow the fundamental physical laws that govern the dynamics of the atmosphere (Watt-Meyer et al., 2023). Additionally, accurately capturing the influence of GHG and aerosols is essential for simulating realistic climate responses to different emission scenarios (Bloch-Johnson et al., 2024).

The few existing climate emulators that incorporate GHG concentrations typically rely on autoregressive training regimes. These models predict climate variables at future time steps based on past states, but often fail to account for the projected emissions at those future times. This limitation leads to significant inaccuracies in predicting future climate states, especially under varying anthropogenic emission scenarios, highlighting a critical gap in current climate modeling approaches.

To address this gap, we propose PACE, which treats climate emulation as a diagnostic-type prediction and integrate emissions data directly into the model's training framework, to predict climate variables from a given parallel time-series of climate forcer emission maps (GHG and aerosols) allowing for more accurate simulation of future climate states under varying concentration scenarios.

Furthermore, we focus on two key phenomenon observed by our climate system i.e. advection and diffusion. In climate modeling, the advection-diffusion equation is fundamental for simulating the transport and dispersion of climate variables, such as temperature and moisture (Choi et al., 2023).

PACE proves to be compute-efficient by modelling key physical law which reduces its dependence on large datasets making it data-efficient as shown in Table 1. Additionally, with generalizability inherent in the advection-diffusion equation, PACE generalizes across 15 climate models emulating surface temperature and precipitation stably for 86 years solely from emissions data (see Figure 1). Our contributions are as follows:

1. We propose PACE, a Neural ODE based climate emulator which models the advection-diffusion phenomenon by dynamically estimating the diffusion coefficient and flow velocities based on the input greenhouse gas concentrations.

2. We introduce Gaussian noise as a stochastic term in advection-diffusion equation to account for uncertainty in climate modelling.

3. We encode periodic boundary conditions by considering the Earth's atmosphere as a spherical domain to faithfully emulate reference climate models.

4. Finally, we perform extensive experiments to show the generalization capabilities of PACE for emulating 15 climate models for 86 years at one time.

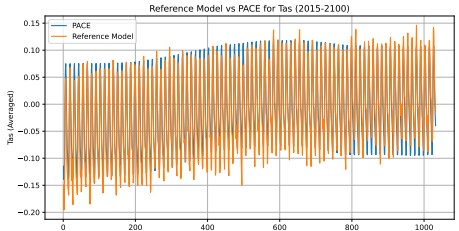 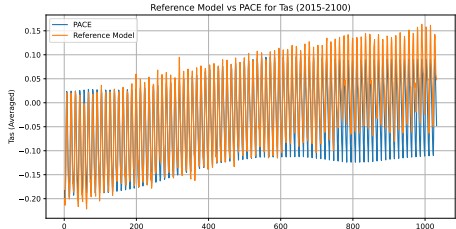

Figure 1: Global averaged Surface Air Temperature (TAS) emulation for 86 years (2015-2100) of Climate Models Left: **FGOALS-f3-L**, Right **TaiESM1**

Table 1: Computational Efficiency of PACE vs several Climate Emulators

| Emulator | Physics Informed | Multi-Model Emulation | Training Resources | Parameters |
|---|---|---|---|---|
| ClimaX | ✗ | ✓ | 32GB NVIDIA V100 | 107M |
| ACE | ✓ | ✗ | N/A | 200M |
| LUCIE | ✓ | ✗ | A100 GPU (2.4hrs) | N/A |
| PACE | ✓ | ✓ | 24GB RTX A5000 | 684K |

## 2 RELATED WORK

### 2.1 MACHINE LEARNING (ML) AND PHYSICS BASED CLIMATE EMULATORS

Recently, ML based and Physics Informed climate emulators have been successful in emulating several climate variables. Watt-Meyer et al. (2023) proposed ACE (AI2 Climate Emulator) based on Spherical Neural Operator (SFNO) architecture for effective physics informed emulation. Guan et al. (2024) proposed LUCIE, also based on SFNO to account for the computational complexity of ACE. Choi et al. (2023) proposed climate modelling using Graph Neural Network (GNN) and Neural ODE, but do not account for GHG emissions or show any long term stability. Additionally, there are several climate emulators which are trained on only one climate model unknown for their generalizability across different climate models (Scher, 2018; Mansfield et al., 2020; Beusch et al., 2020; Cachay et al., 2021; Watson-Parris et al., 2022). Bassetti et al. (2024) use diffusion models for climate emulation, however their primary goal is temporal downscaling. Nguyen et al. (2023a) accounts for multi-model training, however it is limited to medium range forecasting.

## 2.2 Modelling Physical Systems using Neural Networks

The neural ordinary differential equation(ODE) model proposed by Chen et al. (2018) has demonstrated significant potential for solving partial differential equations (PDEs) that govern the complex physical systems, opening up numerous new research avenues in the field (Mattheakis et al., 2022; Dandekar et al., 2020; Finzi et al., 2020; Lutter et al., 2019). Further, physics-informed neural networks (PINNs) were used to to solve the advection-dispersion equation using discretization-free and reduced-order methods (Vadyala et al., 2022; He & Tartakovsky, 2021). Neural Networks (NN) have also been used as surrogate models for obtaining PDE solutions in fluid dynamics and forecasting (Lu et al., 2021; Li et al., 2020; Brandstetter et al., 2022; Sønderby et al., 2020; Keisler, 2022).

## 3 Modelling Climate Variability Through Neural Advection-Diffusion Process

### 3.1 Problem Formulation

We model climate emulation as a continuous sequence to sequence (seq-to-seq) task where the goal is to predict mapping of climate variables from a given time-series of climate forcer emission maps. Considering that climate system evolves according to a 2D advection-diffusion process, described by the following partial differential equation (PDE):

$$\frac{\partial u}{\partial t} + v \cdot \nabla u = D \nabla^2 u \tag{1}$$

where $u(x, y, t)$ represents the climate variables (temperature and precipitation) at time $t$ and spatial coordinates $(x, y)$, $v$ is the velocity field representing advection and $D$ is the diffusion coefficient. Formally, let $\mathbf{F}(t) \in \mathbb{R}^{x \times y}$ represent the input fields of greenhouse gas concentrations at time $t$ and $x$ and $y$ denote the latitude-longitude spatial grid $\in \Omega = [-90°, 90°] \times [-180°, 180°] \subset \mathbb{R}^2$. The output $\mathbf{U}(t) \in \mathbb{R}^{x \times y}$ corresponds to the predicted climate variables at the parallel time step. The neural network is trained to solve the following mapping:

$$\mathbf{U}(t) = \mathcal{M}(\mathbf{F}(t); \theta) \tag{2}$$

where $\mathcal{M}$ is the neural network model parameterized by $\theta$, which approximates the solution to the advection-diffusion equation given the input emissions $\mathbf{F}$. The model is designed to learn the spatiotemporal patterns of our climate system dictated by the underlying physical processes modeled by the PDE. The complete architectural pipeline of PACE is shown in Figure 2

### 3.2 Advection Diffusion Process

We model climate emulation as a continuous spatio-temporal process which captures two fundamental physical processes: advection and diffusion, which together dictate how substances are transported and spread out throughout the climate system. The general form of the advection-diffusion equation in a climate system is defined in equation 1.

To faithfully emulate the climate's chaotic nature, it is essential to determine the path and rate at which the physical quantities are transported given by $v \cdot \nabla u$ where $v$ is the velocity vector of the fluid (e.g., wind velocity) and $\nabla u$ is the gradient of the quantity being transported (e.g., temperature or concentration). On the other hand, diffusion models the distribution of physical quantities such as heat, moisture, and other properties within the atmosphere $D \nabla^2 u$ where $D$ is the diffusion coefficient, indicating how the scalar field spreads out due to molecular diffusion.

We employ Neural ODE presented by Chen et al. (2018) to solve the 2D advection diffusion equation 3 by discretizing the spatial domain using Finite Difference Method (FDM) Fiadeiro & Veronis (1977) considering the earth is divided into spatially uniform grid points in x and y directions (longitude x latitude). FDM employ spatial discretization to approximate derivatives using the values at grid points. We explain the spatail discretization and show it's effect visually in section 4.

$$\frac{\partial u}{\partial t} + v_x \frac{\partial u}{\partial x} + v_y \frac{\partial u}{\partial y} = D(\frac{\partial^2 u}{\partial x^2} + \frac{\partial^2 u}{\partial y^2}) \tag{3}$$

The spatial derivatives are therefore descritized as equation 4, equation 5:

$$\frac{\partial u}{\partial x} \approx \frac{u(x + \Delta x, y, t) - u(x - \Delta x, y, t)}{2\Delta x} \tag{4}$$

$$\frac{\partial^2 u}{\partial x^2} \approx \frac{u(x + \Delta x, y, t) - 2u(x, y, t) + u(x - \Delta x, y, t)}{\Delta x^2} \tag{5}$$

Similarly for $\frac{\partial u}{\partial y}$ and $\frac{\partial^2 u}{\partial y^2}$. The discretized spatial dimensions are substituted in the equation 3, while time t remains continuous. We used dopri5 (Dormand & Prince, 1980) solver to integrate the learned dynamics over time, predicting the evolution of the system as shown in equation 6.

$$\frac{du}{dt} = f_\theta(u, v_x, v_y, D, \Delta x, \Delta y) \tag{6}$$

$$u(t) = dopri5(f_\theta, u_0, t) \tag{7}$$

### 3.2.1 ESTIMATING DIFFUSION COEFFICIENT AND VELOCITY FIELD OF CLIMATE FORCER EMISSIONS

We initialize the model with the empirical estimation of diffusion coefficient D from green house gas emissions data. We calculate the spatial variance across the latitude and longitude dimensions to analyze how greenhouse gas concentrations spread from regions of high emissions over time. The diffusion co-efficient is calculated as equation 8.

$$D_{estimate} = \frac{1}{M} \sum_{i=1}^{M} Var(C_i) \tag{8}$$

where $M$ is the number of gas types and $Var(C) =$ spatial variance calculated as:

$$Var(C) = \frac{1}{N_x N_y} \sum_{x=1}^{N_x} \sum_{y=1}^{N_y} (C(t, x, y) - \bar{C}(t))^2 \tag{9}$$

where $C(t, x, y)$ is the concentration at time $t$ at point $(x, y)$, $\bar{C}(t)$ is the mean concentration across the spatial domain and $N_x, N_y$ are the number of grid points in the longitude and latitude dimensions.

We empirically estimate the initial velocity from GHG concentration fields. The velocity fields $v_x$ and $v_y$ are inferred using spatial gradients of the concentration field as shown in equation 10. These gradients indicate the direction and rate of concentration change, allowing the model to simulate advection accurately. Estimating velocity this way integrates spatial transport dynamics into the advection-diffusion solver, crucial for realistic climate modeling.

$$v_x \approx \frac{\partial C}{\partial x}, v_y \approx \frac{\partial C}{\partial y} \tag{10}$$

$$\frac{\partial C}{\partial x} \approx \frac{C(x + \Delta x, y, t) - C(x - \Delta x, y, t)}{2\Delta x} \tag{11}$$

$$\frac{\partial C}{\partial y} \approx \frac{C(x, y + \Delta y, t) - C(x, y - \Delta y, t)}{2\Delta y} \tag{12}$$

### 3.2.2 UNCERTAINTY ESTIMATION

To account for uncertainty in our climate model, we integrate a stochastic term into the advection-diffusion as show in equation 13. Here, the stochasticity refers to a noise term which represents random fluctuations or uncertainties.

$$\frac{\partial u}{\partial t} + v \cdot \nabla u = D\nabla^2 u + \alpha\eta(x, y, t) \tag{13}$$

where $\eta(x, y, t)$ is a stochastic process, modeled as Gaussian noise with mean zero and variance $\sigma^2$. We control the intensity of the stochastic perturbations with $\alpha$ ass a scaling factor and optimize the Negative Log-Likelihood (NLL) loss which penalizes low probability assignments to observed outcomes. Assuming $y \sim \mathcal{N}(\mu, \sigma^2)$, the loss is given in equation 14

$$NLL = -\frac{1}{N} \sum_{i=1}^{N} [-\frac{(y_i - \mu_i)^2}{2\sigma_i^2} - \log(\sigma_i^2) - \frac{1}{2} \log(2\pi)] \tag{14}$$

where $y$ is the target data and $N = H.W$ product of the height (latitude) and width (longitude) of the spatial grid.

### 3.2.3 Periodic Boundary Conditions and Harmonics Spatio-Temporal Embeddings

We implement periodic boundary condition (PBC) to simulate the entire planet. Considering Earth as roughly spherical, PBC ensure that the boundary at one edge of the domain connects seamlessly to the opposite edge, avoiding artificial edge effects and ensuring continuity. Mathematically, if $f(x, y)$ is the state variable, periodic conditions imply $f(x, y) = f(x + L_x, y) = f(x, y + L_y)$ where $L_x$ and $L_y$ are the domain lengths in the x and y directions, respectively.

We implement harmonic embeddings to learn seasonal variations and cyclical changes in climate data. By employing a series of sine and cosine functions of varying frequencies, these embeddings introduce features that help the model learn and represent periodic behaviors in the data effectively.

$$embedding(t) = [sin(2^i \cdot t), cos(2^i \cdot t), \ldots\ldots, sin(2^{n-1} \cdot t), cos(2^{n-1} \cdot t)] \tag{15}$$

where $n$ is the number of bands and $2^i$ is the frequency factor for each band, where $i$ ranges from 0 to $n - 1$ (determined by maximum frequency).

### 3.3 Convolution Block Attention Module (CBAM)

We implement a CBAM to handle the global spatial dependencies, as a parameterized network equation 16. The Neural ODE models the advection diffusion dynamics and extract features that are then fed into the CBAM which applies both Channel Attention Module (CAM) equation 17 and Spatial Attention Module (SAM) equation 18.

$$f_\theta(u(x, y)) = M_c(F) + M_s(F) \tag{16}$$

$$M_c(F) = \sigma(MLP(AvgPool(F)) + MLP(MaxPool(F))) \tag{17}$$

where $M_c(F)$ is the channel attention map, $\sigma$ is the sigmoid function, and MLP denotes the multilayer perceptron.

$$M_s(F) = \sigma(f^{7 \times 7}([AvgPool(F); MaxPool(F)])) \tag{18}$$

where $M_s(F)$ is the spatial attention map, and $f^{7 \times 7}$ is the convolutional layer with a 7x7 filter.

## 4 Experiments and Results

### 4.1 Task

The goal of PACE is to emulate surface air temperature (TAS) and precipitation (PR) from climate forcer emission maps ($CO_2$, $CH_4$, $SO_2$, BC) for a parallel time series of 2015-2100. We simulate the output of each climate model as single and super emulator, and also validate the generalisation of our methodology using zero-shot learning. We compare PACE against all baselines provided by ClimateSet under the same hyperparameter settings. We also compare against ACE (Watt-Meyer et al., 2023) and LUCIE Guan et al. (2024), two recent climate emulators. Since, they both are developed for different emulation task, we adopt their base architecture SFNO Bonev et al. (2023) and train it for the same task as ours. The details for adaptation of SFNO are given in Appendix A.2.

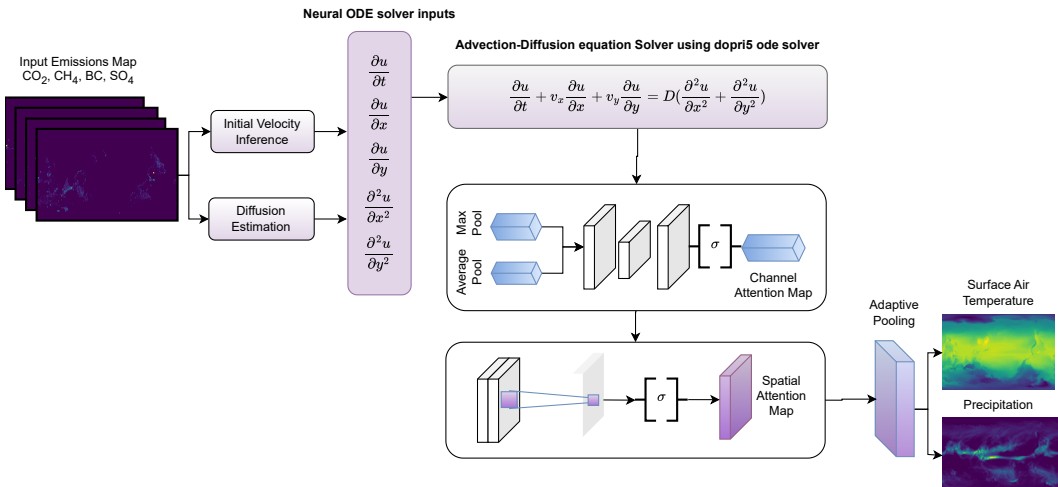

Figure 2: Complete architectural pipeline of PACE. The model initializes the velocity fields and diffusion co-efficient from the input data of four gases i.e. $C0_2, CH_4, BC, SO_2$. After solving the advection-diffusion equation, channel and spatial attention extracts important spatial features to generate output maps of Temperature and Precipitation

## 4.2 DATASET

We train PACE on a total of 15 climate models provided by ClimateSet Kaltenborn et al. (2023). ClimateSet compiles climate data from the Coupled Model Intercomparison Project Phase 6 (CMIP6) (Eyring et al., 2016) , incorporating climate model outputs from ScenarioMIP (O'Neill et al., 2016) and future emission trajectories of climate forcing agents from Input Datasets for Model Intercomparison Projects (Input4MIPs) (Durack et al., 2017). Each climate model has been standardized to a spatial resolution of 250km i.e. $96 \times 144$ grid points (latitude $\times$ longitude) with a monthly temporal resolution. Both input and output datasets consist of 86-year time-series data spanning four SSP scenarios (SSP1-2.6, SSP2-4.5, SSP3-7.0, SSP5-8.5) from 2015 to 2100. We use three scenarios namely SSP1-2.6, SSP3-7.0, SSP5-8.5 for training with a validation split of 0.1 and SSP2-4.5 for testing.

## 4.3 EVALUATION METRICS

We evaluate PACE and all benchmarks using latitude-weighted Root Mean Square Error (RMSE) given in equation 19.

$$RMSE = \frac{1}{N} \sum_{t}^{N} \sqrt{\frac{1}{HW} \sum_{h}^{H} \sum_{w}^{W} L(i)(y_{thw} - pred_{thw})} \qquad (19)$$

where $L(i)$ accounts for latitude weights.

$$L(i) = \frac{\cos(lat(i))}{\frac{1}{H} \sum_{i'=1}^{H} \cos(lat(i'))}$$

where lat(i) represents the latitude of the i-th row within the grid. The latitude weighting factor is introduced to address the uneven distribution of areas when mapping the spherical Earth's surface onto a regular grid.

## 4.4 SINGLE EMULATOR

For single emulator experiments, we trained all models for 25 epochs. We report RMSE for UNet, ConvLSTM, ClimaX, ClimaX_Frozen and SFNO. The training hyperparameters are all kept similar to those used in the original paper. PACE outperforms all models for emulating temperature across

13 climate models whereas ClimaX performs better on EC-Earth3 and TaiESM1 with lowest RMSE. For precipitation emulation, all models perform slightly worse with SFNO having lowest RMSE for simulating BCC-CSM2-MR and TaiESM1, ClimaX for AWI-CM-1-1-MR and CAS-ESM2-0, UNet performs best for INM-CM4-8 while PACE performs best for simulating remaining 10 climate models. Figure 3 shows the overall distribution of how each ML model performs across all 15 climate models in emulating temperature and precipitation. Detailed RMSE results for all 15 climate models for temperature and precipitation emulation are given in Appendix B in Figures 6 and 7.

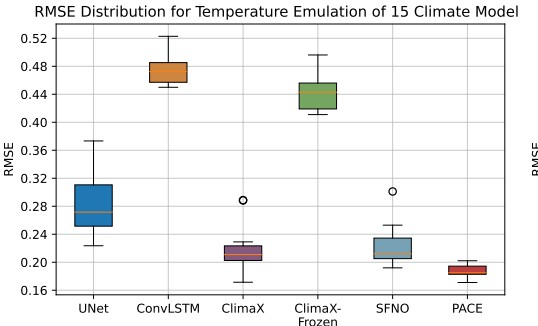 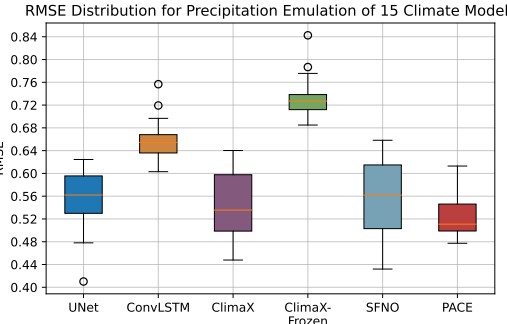

Figure 3: RMSE distribution for climate variable predictions ccross different models. Each box represents the interquartile range (IQR) of RMSE values, capturing the spread and variability in prediction accuracy for key climate variables Temperature and Precipitation. The plot shows the consistent performance of PACE across all 15 climate models.

## 4.5 SUPER EMULATOR

Here, the term super emulator is used to train a single ML model on all of 15 climate models. This leads to the rich feature learning resulting in better generalization capabilities across different climate models. We use the same multihead decoder proposed in ClimateSet to train all ML models including PACE. We train all ML models for 100 epochs to keep the training regime computationally efficient with 2 convolutional layers and and 32 units decoder head. For super-emulator experiments we use a batch size of 1 for all models due to computational constraints.

For super emulation PACE outperforms all ML models on 13 climate models while ConvLSTM performs best for emulating EC-Earth3-Veg-LR and TaiESM1. The authors of ClimateSet Kaltenborn et al. (2023) suggest that during **super emulation**, smaller models demonstrate superior learning efficiency compared to larger models. This is because smaller models converge faster, allowing the model to learn patterns and relationships in the data more rapidly. We believe PACE being physically consistent and compute-efficient is able to learn complex climate features and outperform computationally intensive climate emulators.

## 4.6 GENERALIZATION CAPABILITIES OF PACE AS A SINGLE EMULATOR

We test the generalization capabilities of ML models on three climate models: AWI-CM-1-1-MR, MPI-ESM1-2-HR and FGOALS-f3-L. The RMSE results are shown in Tables 3, 4 and 5 for TAS (surface air temperature) and PR (precipitation) pre-trained on different climate models and tested on these three climate models. The metric for best performing model is emboldened and second best is highlighted in red. The pretrained climate model column shows which dataset the model was initially trained on before being tested on the either of the three climate models. Overall PACE, SFNO and ClimaX generalize well over different climate models with PACE outperforming on majority of the models.

While ClimaX benefits from pre-training on multiple climate models, PACE demonstrates superior computational efficiency and generalization. ClimaX required up to 80 GPUs and a large-scale dataset for pre-training Nguyen et al. (2023a), whereas PACE, pre-trained on a single climate model using just one GPU, achieves superior generalization across diverse climate models. This highlights the resource efficiency of our approach without compromising performance.

Table 2: Super-emulator results on 15 climate models datasets which are a subset of ClimateSet. We report the RMSE for TAS (surface air temperature). The best performing models are emboldened.

| Climate Models | PACE | UNet | ConvLSTM | SFNO | ClimaX | ClimaX$_{frozen}$ |
|---|---|---|---|---|---|---|
| AWI-CM-1-1-MR | **0.259** | 0.406 | 0.322 | 0.666 | 0.896 | 0.780 |
| BCC-CSM2-MR | **0.291** | 0.464 | 0.312 | 0.679 | 0.764 | 0.624 |
| CAMS-CSM1-0 | **0.263** | 0.470 | 0.325 | 0.610 | 0.732 | 0.721 |
| CAS-ESM2-0 | **0.254** | 0.495 | 0.351 | 0.671 | 0.867 | 0.724 |
| CNRM-CM6-1-HR | **0.222** | 0.441 | 0.307 | 0.599 | 0.742 | 0.631 |
| EC-Earth3 | **0.229** | 0.418 | 0.349 | 0.651 | 0.799 | 0.686 |
| EC-Earth3-Veg-LR | 0.297 | 0.398 | **0.278** | 0.589 | 0.756 | 0.701 |
| FGOALS-f3-L | **0.276** | 0.481 | 0.399 | 0.629 | 0.877 | 0.711 |
| GFDL-ESM4 | **0.256** | 0.424 | 0.394 | 0.595 | 0.876 | 0.697 |
| INM-CM4-8 | **0.247** | 0.482 | 0.342 | 0.611 | 0.743 | 0.623 |
| INM-CM5-0 | **0.204** | 0.394 | 0.300 | 0.570 | 0.799 | 0.656 |
| MPI-ESM1-2-HR | **0.245** | 0.430 | 0.337 | 0.673 | 0.801 | 0.767 |
| MRI-ESM2-0 | **0.285** | 0.464 | 0.371 | 0.692 | 0.821 | 0.714 |
| NorESM2-MM | **0.278** | 0.452 | 0.350 | 0.584 | 0.720 | 0.695 |
| TaiESM1 | 0.311 | 0.408 | **0.309** | 0.587 | 0.699 | 0.617 |

Table 3: Generalization results on AWI-CM-1-1-MR. The first row shows the results from training on AWI-CM-1-1-MR from scratch.

| Pre-Trained Climate Model | PACE | | UNet | | ConvLSTM | | SFNO | | ClimaX | | ClimaX$_{frozen}$ | |
|---|---|---|---|---|---|---|---|---|---|---|---|---|
| | TAS | PR | TAS | PR | TAS | PR | TAS | PR | TAS | PR | TAS | PR |
| AWI-CM-1-1-MR | **0.184** | 0.499 | 0.289 | 0.571 | 0.451 | 0.622 | 0.200 | 0.501 | 0.207 | **0.498** | 0.412 | 0.707 |
| BCC-CSM2-MR | 0.247 | 0.620 | 0.275 | 0.653 | 0.466 | 0.696 | 0.230 | **0.599** | 0.232 | 0.617 | 0.432 | 0.753 |
| CAS-ESM2-0 | 0.275 | **0.641** | 0.273 | 0.694 | 0.477 | 0.714 | 0.278 | 0.656 | **0.250** | 0.654 | 0.461 | 0.763 |
| MRI-ESM2-0 | **0.205** | **0.620** | 0.276 | 0.656 | 0.456 | 0.697 | 0.218 | 0.635 | 0.223 | 0.643 | 0.410 | 0.756 |
| NorESM2-MM | **0.231** | 0.582 | 0.301 | 0.562 | 0.459 | 0.674 | 0.245 | 0.591 | 0.286 | **0.551** | 0.441 | 0.754 |

Table 4: Finetuning results on MPI-ESM1-2-HR. The first row shows the results from training on MPI-ESM1-2-HR from scratch.

| Pre-Trained Climate Model | PACE | | UNet | | ConvLSTM | | SFNO | | ClimaX | | ClimaX$_{frozen}$ | |
|---|---|---|---|---|---|---|---|---|---|---|---|---|
| | TAS | PR | TAS | PR | TAS | PR | TAS | PR | TAS | PR | TAS | PR |
| MPI-ESM1-2-HR | **0.193** | 0.477 | 0.234 | **0.410** | 0.449 | 0.636 | 0.198 | 0.586 | 0.214 | 0.509 | 0.411 | 0.716 |
| CNRM-CM6-1-HR | **0.221** | 0.580 | 0.268 | 0.657 | 0.482 | 0.714 | 0.234 | 0.601 | 0.225 | 0.599 | 0.453 | 0.756 |
| EC-Earth3 | 0.228 | **0.544** | 0.262 | 0.558 | 0.465 | 0.667 | 0.259 | 0.551 | **0.220** | 0.567 | 0.443 | 0.744 |
| EC-Earth3-Veg-LR | 0.241 | 0.554 | 0.270 | 0.560 | 0.471 | 0.660 | 0.244 | 0.565 | **0.233** | **0.551** | 0.456 | 0.744 |
| TaiESM1 | **0.208** | **0.602** | 0.294 | 0.711 | 0.461 | 0.692 | 0.248 | 0.645 | 0.268 | 0.682 | 0.427 | 0.757 |

Table 5: Finetuning results on FGOALS-f3-L. The first row shows the results from training on FGOALS-f3-L from scratch.

| Pre-Trained Climate Model | PACE | | UNet | | ConvLSTM | | SFNO | | ClimaX | | ClimaX$_{frozen}$ | |
|---|---|---|---|---|---|---|---|---|---|---|---|---|
| | TAS | PR | TAS | PR | TAS | PR | TAS | PR | TAS | PR | TAS | PR |
| FGOALS-f3-L | **0.184** | **0.559** | 0.241 | 0.562 | 0.485 | 0.652 | 0.253 | 0.561 | 0.218 | 0.573 | 0.456 | 0.729 |
| GFDL-ESM4 | 0.207 | **0.563** | 0.321 | 0.716 | 0.484 | 0.697 | 0.271 | 0.600 | 0.330 | 0.708 | 0.468 | 0.762 |
| INM-CM4-8 | 0.232 | **0.722** | 0.296 | 0.776 | 0.491 | 0.744 | 0.259 | 0.737 | 0.245 | 0.745 | 0.468 | 0.790 |
| INM-CM5-0 | 0.212 | **0.574** | 0.277 | 0.756 | 0.488 | 0.739 | 0.261 | 0.730 | 0.250 | 0.725 | 0.459 | 0.785 |
| MPI-ESM1-2-HR | **0.209** | 0.696 | 0.283 | 0.701 | 0.482 | 0.714 | 0.257 | 0.701 | 0.250 | **0.691** | 0.445 | 0.760 |

## 5 NUMERICAL DISCRETIZATION AND GRID REPRESENTATION: IMPACT OF FINITE DIFFERENCE METHODS (FDM) ON EARTH'S SPATIAL GRIDDING IN CLIMATE MODELS

We utilize FDM for spatial discretization in PACE which divides the the physical space (in this case Earth atmosphere) into a grid of discrete points. Each grid point represents a specific location, and the value of the physical quantity (e.g., temperature) is computed at each point. The gridding at a lower resolution does induce additional errors. In future, we aim to test FVM and FEM to test if they results in smoother outputs and reduce errors.

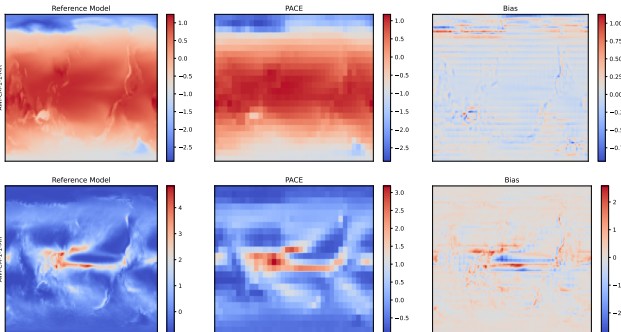

Figure 4: Numerical descrization effect on emulation of Temperature and Precipitation. In FDM, continuous differential equations (like the equations governing temperature, pressure, or velocity) are approximated using discrete differences between values at specific grid points. The process of discretization converts the continuous space into a finite grid, and the differential operators (like derivatives) are approximated using differences between the values at neighboring grid points.

## 6 ABLATION STUDIES

To understand the importance of each component of PACE, we perform ablation studies across four climate models namely AWI-CM-1-1-MR, TaiESM1, EC-Earth3 and NorESM2-MM.

**Advection-Only:** For this study, we remove the empirical estimated diffusion term from PACE and only model the advection process using Neural ODE. The resulting RMSE for surface air temperature and precipitation increases deteriorating the model's overall performance. The results show that missing approximation of diffusion has a greater effect on temperature as compared to precipitation, therefore determining that diffusion is critical in accurately simulating the transport of physical quantities like heat, moisture, and momentum.

**Diffusion-Only:** In order to understand the importance of advection process, we initialised the model with constant velocities i.e. ($v_x = 1.0, v_y = 1.0$) rather than estimating their values from GHG emissions. This resulted in a much greater impact on emulating precipitation as compared to the previous study.

**Neural ODE:** For this study, we remove the advection diffusion process and only parameterize the Convolution Attention Module using Neural ODE. Our results demonstrate that accurately capturing advection-diffusion process is essential to simulate how energy and moisture are distributed, which directly impacts predictions of temperature, precipitation, and long-term climate changes, highlighting the critical contribution of each element to optimizing the model's emulating performance.

## 7 CONCLUSION AND FUTURE WORK

In this work, we present PACE, a physics and uncertainty aware climate emulator which accounts for Earth's atmospheric advection-diffusion phenomenon. We incorporate a key physical law in PACE by solving a time-dependent partial different equation (PDE) using Neural ODE. Additionally, we encode periodic boundary conditions to avoid artificial edge effects that arise from rigid boundaries.

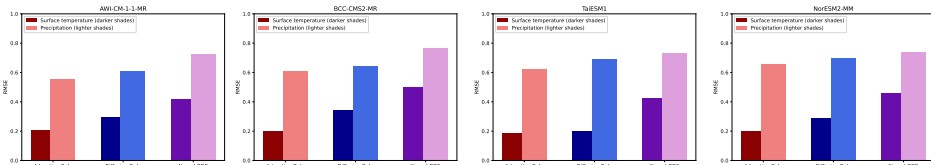

Figure 5: Ablation studies for TAS and PR emulation. Advection (red), diffusion (blue) and NODE (purple). We can see that advection plays the most important role in emulation followed by diffusion.

However, there are a few potential limitations of PACE which will be addressed as a possible future work. Our training regime only includes climate model data which is based on simulations and not considered entirely accurate. As a future work, we aim to extend training mechanism of PACE on both ERA5 weather and ClimateSet's extensive climate data to enhance emulation accuracy. Additionally, PACE is trained on coarse resolution data which does not fully account for extreme events at regional level. Further, PACE is still limited in its ability to emulate precipitation stably for 86 years. These limitations can be addressed by training on high resolution data and encoding physical constraints such as energy, mass and water conservation in a loss function.

## ETHICAL STATEMENT

Our research aims to emulate temperature and precipitation for multiple climate models by solving an atmospheric advection-diffusion equation using ML based approach while being computationally efficient. The findings demonstrate that data-driven approaches can substantially enhance forecast accuracy while utilizing computational resources more efficiently. The environmental impact of optimizing computational efficiency in forecasting is notable, as it reduces the carbon footprint associated with large-scale computational processes, aligning with global initiatives to mitigate climate change. By combining machine learning (ML) techniques to both improve predictive accuracy and reduce computational overhead, we propose a sustainable and scalable solution for climate emulation that can better serve the global population.

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

# A  EXPERIMENT DETAILS

## A.1  HYPERPARAMETERS

Table 6: Training Hyperparameters for PACE

| Hyperparameters | Meaning | Value |
|---|---|---|
| in_var | Number of input variables | 4 |
| out_var | Number of output variables | 2 |
| solver_method | Numerical integration | dopri5 |
| rtol | Relative tolerance | 1e-3 |
| atol | Absolute tolerance | 1e-6 |
| num_bands | Number of frequency bands | 4 |
| max_freq | Maximum frequency for harmonic embeddings | 6 |
| batch_size | Batch size | 4 |
| optim | Optimizer | Adam |
| lr | Learning rate | 2e-4 |
| $lr_{scheduler}$ | Learning Rate Scheduler | Exponential |
| decay | Weight decay value | 1e-4 |
| $\epsilon$ | epsilon value | 1e-8 |
| norm | Data Normalization | z-score |

Table 7: Hyperparameters for Convolutional Block

| Hyperparameters | Meaning | Value |
|---|---|---|
| conv2d | Number of convolutional layers | 4 |
| hidden_channels | Number of hidden layers | 64 |
| channel increment | Multiplication factor for hidden layers | [1,2,2,4] |
| kernel_size | Convolution filter size | 3 |
| stride | Stride of each convolution layer | 1 |
| padding | Padding of each convolution layer | 1 |
| cbam | Number of CBAM layers | 3 |
| activation | Activation Function | ReLU |
| dropout | Dropout rate | 0.1 |

## A.2  SFNO TRAINING DETAILS

We maintain the hyperparameters of the SFNO consistent with the configuration proposed in LUCIE (Guan et al., 2024). To adapt SFNO for our specific task, we incorporate a 2D convolutional layer designed to handle inputs with 4 channels and produce outputs with 2 channels. This modification ensures compatibility between the original model architecture and the dimensional requirements of our data, allowing effective processing of our input-output pair while maintaining the integrity of the model's core hyperparameters.

Table 8: Hyperparameters for SFNO

| Hyperparameters | Value |
|---|---|
| SFNO blocks | 6 |
| Encoder and Decoder Layers | 1 |
| Units per Layer | 32 |
| Optimizer | Adam |
| Learning Rate | $1 \times 10^{-4}$ |
| Activation Function | GELU |

## A.3 HARDWARE AND SOFTWARE REQUIREMENTS

We use PyTorch Paszke et al. (2019), Pytorch Lightning Falcon (2019), torchdiffeq Chen et al. (2018) for implementation of PACE. We train PACE on a single RTXA5000 with 24GB RAM. We perform all super emulator training experiments on a single NVIDIA DGX A100 with 80 GB RAM.

## B RESULTS

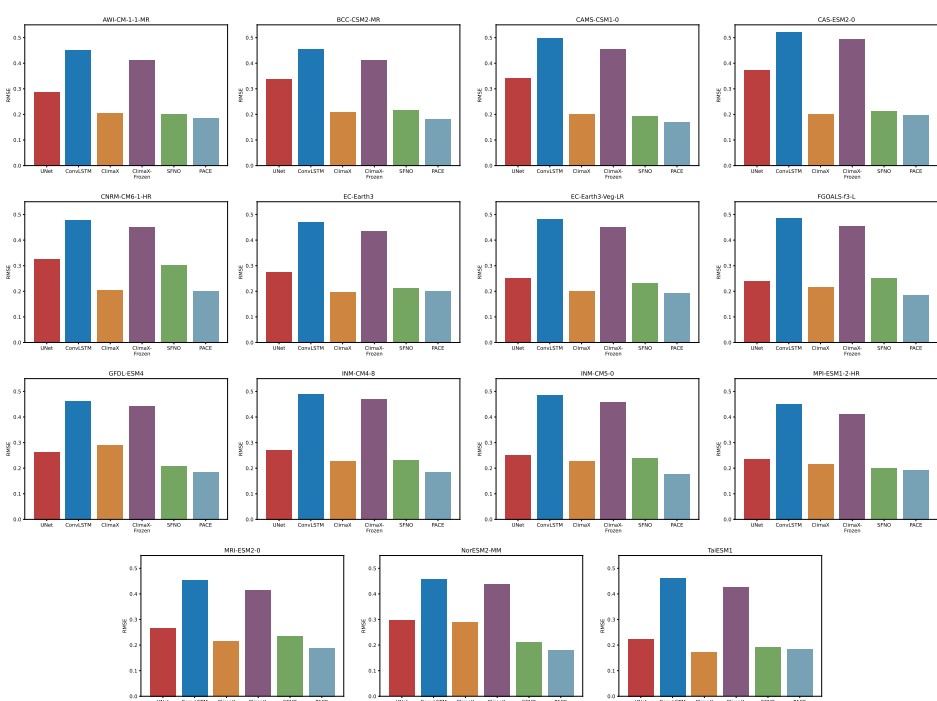

Figure 6: RMSE results for Surface Air Temperature (TAS) Emulation for the projection of SSP2-4.5 (2015 − 2100). PACE outperforms all other ML models on 13 out of 15 Climate models namely:

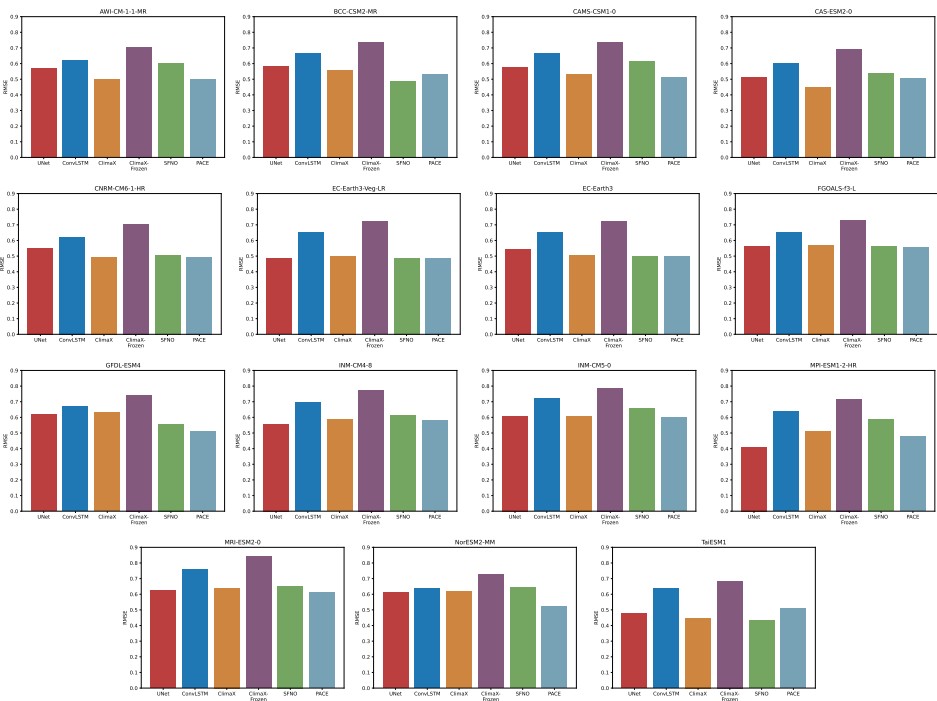

Figure 7: RMSE results for Precipitation (PR) Emulation for the projection of SSP2-4.5 (2015 – 2100). PACE outperforms all other ML models on 9 out of 15 Climate models namely:

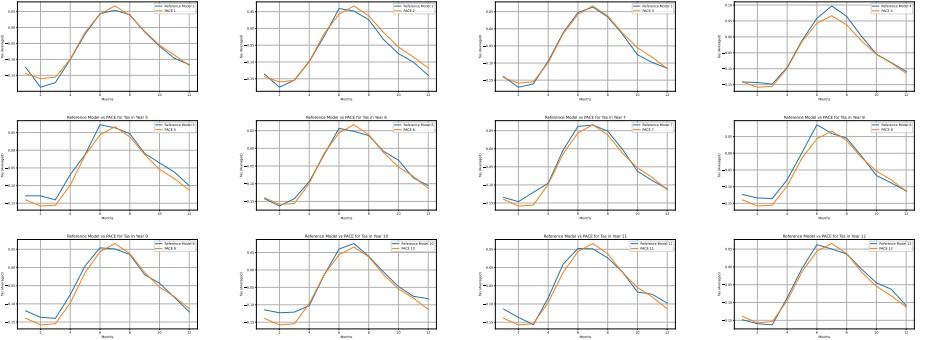

Figure 8: Temperature emulation of the year 2015-2026 for Climate Model AWI-CM-1-1-MR