# OpenReview forum: "PACE: Physics Informed Uncertainty Aware Climate Emulator"
_ICLR.cc/2025/Conference — ICLR 2025 Conference Withdrawn Submission_

### Official Review · Reviewer_qQui · 2024-10-22

**Soundness:** 2
**Presentation:** 1
**Contribution:** 3
**Rating:** 1
**Confidence:** 3

**Summary:**

The paper proposes an ODE/PDE-based climate simulator trained from climate models. The methodological contribution is adding diffusion and noise process to the dynamics.

**Strengths:**

- The paper boasts outstanding results.
- The idea of adding diffusion, noise and greenhouse forcings to advection ODEs is excellent.

**Weaknesses:**

- The language of the paper is weak, and hurts the paper’s clarity and message. Many sentences are oddly phrased, broken or difficult to understand. The language is imprecise and lacks citations. Notation is imprecise, there are hanging equations, poor formatting, etc. The figures are poorly formatted. Experiments and results are difficult to follow.
- I did not really understand what this paper even does. I think the model learns to match to output of existing climate simulator. I didn’t understand how the ODE operates, and what the neural networks are doing. I think the paper is claiming to do 86 year long ODE rollout, but I have really hard time believing this with pretty much all details missing. If this is really done, how can it be in any way accurate? Where are the GHG emissions coming from for 2090..?
- The method contributions of noise process, diffusion constant estimation, and boundary conditions are unconvincing. The noise process is not presented as proper Wiener, and I don’t see how it doesn’t lead to explosion or vanish. The constant estimation is lacks motivation, and makes little sense. I don’t think boundary conditions are implemented, despite claims.

**Questions:**

- Eq 1 doesn’t have divergence. Is this intentional? There are also no sinks or sources, so this system is limited to div-free dynamics. Is climate div-free? There are also forcings or GHG here. f_theta is undefined.
- I don’t understand the overall dynamical system. The eq 1 says that we use an advection-diffusion equation. Ok, but then eq 6 says that we use a neural network instead of that (I assume). And then eq 2 says that we also use another neural network to predict states from GHG. I’m just lost on what happens. Having an algorithm box would, or complete model description would help. Fig2 isn’t helpful (quite the opposite).
- Why is D chosen as var(C)? I see no motivation, and no reason why it should be like this. I don’t really see a connection from GHG to diffusion in the first place. The GHG adds energy to the system, which should amplify the advection. Instead, in this model extra emissions are smoothing the weather.
- I don’t see why the embeddings would help with the boundary conditions. The state can still have arbitrary values at the borders.
- I did not understand the experimental setting: what do you do, what is data, what is known, what is unknown, what do you estimate, what do you predict, what do you simulate. I think that you are mapping emission inputs into temperatures, separately for each time snapshots. Some ODE simulations are inside this, but not sure what.

---

### Official Review · Reviewer_D1xx · 2024-10-28

**Soundness:** 2
**Presentation:** 1
**Contribution:** 2
**Rating:** 3
**Confidence:** 4

**Summary:**

The paper presents a physics-informed approach to climate modeling via incorporating a stochastic noise term for uncertainty estimation within the general advection-diffusion equation. By integrating physical priors, the method enhances benchmark metrics across a range of tasks.

**Strengths:**

- Incorporation of physical priors for climate modeling
- Improvements in downstream performance over various tasks

**Weaknesses:**

- The paper is not well written.
- The paper misses out on a ton of related works which resolve around the same idea of utilizing physics priors to model climate and weather, such as ClimODE (https://arxiv.org/abs/2404.10024), Neural GCM (https://arxiv.org/abs/2311.07222), WeatherGFT (https://arxiv.org/pdf/2405.13796), etc all seem to be missing and comparisons to them are missing.
- Since, the forecasts are probabilistic (uncertainty), it is much better to use CRPS scores to compare to the ground truth as compared to RMSE

**Questions:**

- Do you dynamically update the velocity predictions ($v_x$, $v_y$), or are they solely based on the initial estimates from the data?
- Can you clarify how you derive uncertainty estimates? It appears from Eq. 6 and 7 that you use $f_\theta$ to evolve the system and obtain forecasts for the advection-diffusion equation, but then a stochastic term appears in Eq. 13. How does this fit together?
- Do you forecast the evolution of the forcing variables?
- Could you explain how you implement periodic boundary conditions? I found limited information in Section 3.2.3. Additionally, how did you determine the values of $L_x,L_y$ for the convolutional block?
- For how many time-points you unroll the trajectory to obtain the solution and forecasts, and how many time-points do you use to estimate the initial velocity?
- There is a missing square in Eq.19.

---

### Official Review · Reviewer_mPNq · 2024-10-31

**Soundness:** 2
**Presentation:** 1
**Contribution:** 2
**Rating:** 3
**Confidence:** 5

**Summary:**

The paper proposes a hybrid physics- and machine-learning-based climate emulator model called PACE. The model combines numerical solving of the advection-diffusion equation with convolutional deep learning components to produce prediction of surface temperature and precipitation. The model takes as input at each time greenhouse gas emissions, and is tasked with learning their impact on the atmosphere. The authors experiment with using the PACE model to emulate multiple climate models, both independently and with one learned super emulator for multiple climate models. PACE achieves lower RMSE values for most climate models compared to baselines.

**Strengths:**

1. Using machine learning to learn climate emulators is a highly interesting problem with potential for large impact. The authors have accurately identified that there are many open questions in the area, and in particular the ability of the machine learning models to incorporate more complex forcing from greenhouse gas emissions is a key property that needs further development.
2. The use of the hybrid approach with the advection-diffusion equations seems well-motivated and leads to an impressively stable model. The authors show stable rollouts for 86 years.
3. Compared to baseline models PACE achieves lower RMSEs for single-model evaluation. Also when training PACE as a super-emulator the errors achieved are generally lower than baselines.
4. A valuable ablation study is performed in section 6. This study clearly shows that all parts of the model are important for the performance of the model.

**Weaknesses:**

1. The presentation of the model throughout section 3 is incoherent and hard to understand. Because of this it is challenging to really understand how the different parts of the model fit together and the motivation behind them. There are different variables used in the different subsections and no clarifications on how they relate. Particularly unclear is:
    * What exactly is the shape of $F$ in section 3.3, and what does it contain specifically? How does this relate to each $u$ being modeled in section 3.1?
    * Section 3.2.1 explains how $D$, $v_x$ and $v_y$ are initialized, but after time 0 where do these values come from? While maybe the diffusion coefficient $D$ could be considered fixed, it is unclear where $v_x$ and $v_y$ come from in eq. 6 past the initial time.
    * The description of the deep learning components of the model in section 3.3 is unclear. The dimensions of $F$ are never given, and it is ambiguous which dimensions the different components operate over (e.g. the pooling and MLPs). These components are described as "attention map/module", but this is not any form of attention mechanism.
2. The authors list as one of the main contributions of the work that the model respects the spherical geometry of the earth through periodic boundary conditions. However, periodic boundary conditions as described do not treat the earth as a sphere, but as a torus. Periodic boundary conditions are reasonable in the longitude direction, but not for latitude. Consider specifically the description in section 3.2.3, where this condition is described as $f(x, y) = f(x, y +L_x)$ (assuming here latitude is $y$). If we set $y = -90^{\circ}$ (the south pole) and $L_y = 180^{\circ}$ (length of the domain), then this becomes $f(x, -90^{\circ}) = f(x, 90^{\circ})$. This means that the values at the south pole are enforced to be the same as at the north pole, which is clearly not desirable.
3. Listed as a main contribution of the paper, and even in the title, is the fact that the method is uncertainty aware. The uncertainty estimation presented is however very unclear. Specifically:
    * The uncertainty estimation part of the model, presented in section 3.2.2 leaves more questions than answers, making the soundness of this approach hard to judge. There is some Gaussian process $\eta$ being added to the advection-diffusion equation. What is the covariance structure of this process?
    * The model is trained with an NLL loss, but it is entirely unclear where the $\mu$ and $\sigma$ going into this loss come from. Going by notation, the $\sigma$ is the same as for the Gaussian process, but this would be very strange as that is not the standard  deviation related to any prediction. Or should this be understood as sampling an ensemble prediction from the model, using different realizations of $\eta$, and then estimating $\mu$ and $\sigma$ from these samples and computing the NLL loss? In any of these cases the Gaussian assumption seems overly simplistic, capturing only the first two modes of the distribution.
    * The capabilities of the model to accurately capture uncertainty are never evaluated in any experiments, as only RMSE is considered. This practically nullifies this contribution, as there is nothing in the paper that supports claims that the model is uncertainty aware.
4. It is questionable if RMSE w.r.t. the actual state of the atmosphere (temperature and precipitation) really is particularly interesting for a climate emulator. In climate modeling we generally do not care about exactly predicting the temperature on some specific day, as we are far beyond any predictability limits. Instead we care about capturing overall statistics and trends. So while a model might have very poor RMSE for every day, it might still be a useful climate model if it accurately captures the general climate (statistics of the temperature). Other works, such as Watt-Meyer et al. (2023) consider e.g. RMSE of temporal means, which seems to be of greater interest. The RMSE used in this paper is more akin to the RMSE computation used in weather forecasting, which is a different problem.

**Minor:**
1. I am missing results for the full model for comparison in figure 5.
2. In the abstract the authors write "PACE emulates temperature and precipitation stably for 86 years while only being trained on greenhouse gas emissions data.". This is inaccurate, as the model clearly uses temperature and precipitation data for training as well.
3. Claims in the 4th paragraph in section 1 should be backed up by references.
4. There are a few highly related works that I miss a proper discussion of in the related work section: (Salva Rühling, et al., 2024), (Kochkov, et al., 2024). I would like to know how the proposed method relates and potentially differs to these.
5. For any description of the baseline models and the multihead decoder used in section 4.5 the authors rely fully on referencing ClimateSet (Kaltenborn et al., 2023). This makes the paper not feel self-contained and the experiments hard to follow without cross-checking the ClimateSet paper.
6. Equations should be properly typeset with non-variables in text font rather than math font, e.g. \text{MLP} instead of $MLP$.
7. Text in plots in the paper is so small that it is nearly impossible to read.

*References:*

* Watt-Meyer, Oliver, et al. "ACE: A fast, skillful learned global atmospheric model for climate prediction." arXiv preprint arXiv:2310.02074 (2023).
* Cachay, Salva Rühling, et al. "Probabilistic Emulation of a Global Climate Model with Spherical DYffusion." arXiv preprint arXiv:2406.14798 (2024).
* Kochkov, D., Yuval, J., Langmore, I. et al. Neural general circulation models for weather and climate. Nature 632, 1060–1066 (2024).
* Kaltenborn, Julia, et al. "Climateset: A large-scale climate model dataset for machine learning." Advances in Neural Information Processing Systems 36 (2023): 21757-21792.

**Questions:**

1. What is the time step used in the numerical solver? What is the temporal resolution of the data?

My other questions are for context intertwined with the points under weaknesses listed above.

---

### Official Review · Reviewer_gcZ7 · 2024-11-02

**Soundness:** 1
**Presentation:** 3
**Contribution:** 2
**Rating:** 3
**Confidence:** 4

**Summary:**

The authors present a novel model for regressing climate responses on to anthropogenic emissions, utilizing a 2-D advection diffusion based NeuralODE. They apply the model to the ClimateSet dataset and compare with appropriate baselines, finding improved accuracy for their approach.

**Strengths:**

The authors use a standard dataset and compare against appropriate baselines. They include an ablation study which explores the contribution of their skill to different aspects of their model. The model appears to achieve good skill in comparison to these baselines, although I have serious misgivings about the approach and its motivation (outlined below).

**Weaknesses:**

This paper has a number of serious, and some more minor weaknesses which I will outline below.

Major weaknesses:
1) The authors use a 2-D advection diffusion model as the basis for their NeuralODE "considering that climate system evolves according to a 2D advection-diffusion process", this is not true to any sensible approximation. Over weather time scales the atmosphere evolves according to Navier Stokes, and over the longer timescales described here there is minimal largescale advection. There is certainly no diffusion in the emissions (c.f. section 3.2.1). The emissions are prescribed from monthly varying estimates of large-scale industrial sources - any changes in location or magnitude are purely socio-economic, there is no effect of the atmosphere. Similarly, while the pattern of the temperature and precip emerge from Navier-Stokes over long time averages, the changes on monthly timescales are almost entirely due to changing boundary conditions. This undermines their rationale for the NeuralODE and renders their estimate of the coefficients in 3.2.1 as nonsense. It also represents a fundamental misunderstanding of the dataset and the task.

2) The authors claim to frame climate emulation in a new setting due to inadequacies in autoregressive models (L34-51). This is not true. Their baseline dataset, ClimateSet uses such a framing, and is itself based on ClimateBench (Watson-Parris et al. 2022) for which there are many example emulators (such as Bouabid et al. 2024), and is itself is only an extension of years of literature in framing climate model emulation in this way (e.g. Castruccio et al., 2014; Holden & Edwards, 2010). Discussion around and claims of 'stability' (L17, L57, L103) are also nonsense, as these are regression based models and stable by construction.

3) The discussion around periodic boundary conditions (3.2.3) is very confusing. The authors start by discussing the spatial periodicity of the Earth given the spherical surface they're modeling, but then model that using 'harmonic embeddings to learn seasonal variations and cyclical changes in climate data.' which models a different kind of periodicity that is inherently temporal.

4) The evaluation of the models against the full 2015-2100 period of SSP245 is misguided, and the comparison against the first 10 years (shown in Figure 8) is wrong. The original ClimateBench (Watson-Parris et al. 2022) protocol which ClimateSet is based on, evaluates against the last 20 years of SSP245 because the first ~20-50 years are very similar to the other scenarios used for training and therefore not a good metric of skill.

Given the fundamental lack of understanding of the problem space shown by the authors (based on the above), I have little faith that the comparison models (ACE and LUCIE) have been faithfully transferred to their task and therefore have little faith in their comparisons. I also find it unlikely that the ConvLSTM performs worse than a UNet if they are of comparable size (but no details are provided for such a comparison). If the skill presented by the authors is real, I suspect it stems from their modeling of uncertainty and the (unphysical) regularization provided by the NeuralODE to avoid overfitting.

References:
 - Watson-Parris, D., Rao, Y., Olivié, D., Seland, Ø., Nowack, P., Camps-Valls, G., et al. (2022). ClimateBench v1.0: A benchmark for data-driven climate projections. Journal of Advances in Modeling Earth Systems, 14, e2021MS002954. https://doi.org/10.1029/2021MS002954
 - Bouabid, S., Sejdinovic, D., & Watson-Parris, D. (2024). FaIRGP: A Bayesian energy balance model for surface temperatures emulation. Journal of Advances in Modeling Earth Systems, 16, e2023MS003926. https://doi.org/10.1029/2023MS003926
 - Castruccio, S., McInerney, D. J., Stein, M. L., Crouch, F. L., Jacob, R. L., & Moyer, E. J. (2014). Statistical emulation of climate model projections based on precomputed GCM runs. Journal of Climate, 27(5), 1829–1844. https://doi.org/10.1175/jcli-d-13-00099.1
 - Holden, P. B., & Edwards, N. R. (2010). Dimensionally reduced emulation of an AOGCM for application to integrated assessment modelling: Dimensionally reduced AOGCM emulation. Geophysical Research Letters, 37(21). https://doi.org/10.1029/2010gl045137

**Questions:**

The authors are welcome to respond to the criticisms laid out above.

---

### Official Review · Reviewer_N16f · 2024-11-02

**Soundness:** 2
**Presentation:** 1
**Contribution:** 2
**Rating:** 3
**Confidence:** 4

**Summary:**

The paper introduces a light-weight, physics-informed neural network, PACE, for learning the mapping from forcing scenarios (e.g. greenhouse gas concentrations; GHG) to atmospheric states (here, temperature and precipitation) in a diagnostic-type approach for climate model emulation.
PACE uses physics-informed features, extracted by a Neural ODE that solves the 2D advection-diffusion equation for the GHG global inputs maps, to predict the corresponding global temperature and precipitation maps for the given forcings timestep (month and year) with a lightweight neural block based on convolutions, pooling layers, and MLPs.
PACE achieves promising RMSE results on the ClimateSet benchmark dataset when compared to other neural network-based emulators.

**Strengths:**

- Efficient, accurate climate emulation is an important topic with the potential for significant impact in democratizing cheap climate projections under various greenhouse gas emission scenarios.
- Using a physics-informed featurization technique based on an advection-diffusion equation is an original idea.
- PACE generally is amongst the best, if not the best, performing models compared to the reported baselines, in terms of RMSE.

**Weaknesses:**

1. Grammatical errors and careless statements plague the manuscript. It should be carefully proofread. I'm including some of the grammar mistakes/typos at the end of the "Weaknesses" section. Here are some examples for the careless statements, just from the introduction and related work:
- "The past decade has seen superior performing data-driven weather forecasting models" -> "The past **years** have seen superior performing data-driven weather forecasting models"
- The sentence *"the medium range forecasting ability makes them unstable for climate modelling several years into the future"* doesn't make sense. Just because a model is able to perform medium-range forecasting doesn't make it unstable (see e.g. ACE).
- Citing Gupta & Brandstetter (2022) in the context of *"Climate models are governed by temporal partial differential equations (PDEs)"* doesn't make sense. If you choose to cite something, it would be better to cite a standard textbook on climate science or at least a climate science paper.
- I don't think that Table 1 shows anything related to data-efficiency, as claimed by the authors *"making it data-efficient as shown in Table 1"*. The authors might have meant computational efficiency.
- The claim that *"To address this gap, we propose PACE, which treats climate emulation as a diagnostic-type prediction"* is misleading without making clear that prior work (e.g. ClimateBench or ClimateSet) does exactly this.
- I don't think that *"Nguyen et al. (2023a) accounts for multi-model training, however it is limited to medium range forecasting."* is true. ClimaX contains climate emulation experiments on the ClimateBench dataset.
- The citation format is sometimes off. Not using brackets for non-inline citations hurts the reading flow.

2. Basic mistakes or imprecisions:
- Including ACE and LUCIE in Table 1 is unfair since they were designed for the "autoregressive" climate-dynamics emulation problem rather than the diagnostics-type emulation problem studied in this paper. The inputs-outputs are quite different between these emulation approaches. The relationship between them is more complex in the autoregressive case.
- Section 3.3.: The name of the Convolution Block Attention Module (CBAM) is misleading since it contains no attention layers. Similarly for the "channel attention map" and the "spatial attention map".
- The abstract mentions that *'While deep learning methodologies have made significant progress in weather forecasting, they are still unstable for climate emulation tasks"*. In my opinion, this statement is wrong and misleading: i) ACE, LUCIE, or Spherical DYffusion [1] are counterexamples of pure deep learning methods that perform stable climate long-term climate emulation with reasonable weather forecasting skill; ii) The statement suggest to me that the paper deals with emulation of ***temporal*** climate dynamics (and producing stable, long-term rollouts). However, this is not true since the paper deals with diagnostic-type climate emulation where the mapping from forcings (e.g. GHG) to climate states (e.g. temperatures) are learned (climate dynamics are not being emulated).
 - Adaptation of SFNO architecture (especially Appendix A.2) is not consistent with the configuration from LUCIE (nor is it with the one from ACE nor the original SFNO paper) as wrongly claimed by the paper. For example, the latent dimension is 72 for LUCIE and 256 for ACE, which are both much larger than the 32 used in this paper (similarly for the number of SFNO blocks). Lastly, it's not clear to me why the paper chooses to add *"a 2D convolutional layer designed to handle inputs with 4 channels and produce outputs with 2 channels"* rather than simply changing the number of input and output channels of the original SFNO architecture.

3. The strength of the results is debatable
- I have doubts about the interpretation shown in Fig. 1. The climate models show clear increasing temperature trends, which are not properly emulated by PACE. In one case there's no clear increasing trend (is PACE simply learning the mean?), in the other case it's much smaller than the climate model one. As a side question, what SSP is this? Can you include that in the caption please?
- Fig. 4 shows that PACE's predictions are very pixelated. This is a problem in climate modeling, where high spatial resolutions are highly desirable. The climate models in CMIP6 are already relatively coarse, so it seems important to at least keep their granularity.
- No error bars are included. I strongly recommend re-training PACE (and the best baselines) with different random seeds, and reporting error bars on the corresponding RMSEs. Otherwise, it is hard to judge how significant the results are, especially since the main results (e.g. Fig. 3, 6, 7) don't seem to indicate a clear edge for PACE compared to the baselines.
- Diagnostic-type climate emulation, as studied in this paper, of temperature (and in some cases even for precipitation [2]) has been shown to work well with simple, non-neural ML approaches like Gaussian Processes (see ClimateBench and ClimateSet) and even linear regression (see [2]). Including these approaches would be crucial, given their simplicity. I appreciate the point of the authors that achieving good RMSEs on ClimateSet with a lightweight neural network is possible, but these non-neural approaches are important to include to carefully compare PACE to even more lightweight approaches.
- The title and model mention uncertainty aware climate emulation, but none of the experiments study this (e.g. ensembling and comparisons to the CMIP6 ensembles themselves).
- Can you share insights with respect to the training and inference runtimes of PACE? How does it compare to fully neural approaches that don't require ODE solvers?

4. Some method details are unclearly presented/lack explanation.
- Can you elaborate on Eq. 14? The relationship to Eq. 13 and the rest of the paper is not clear to me. Is $y$ the climate model temperature/precip. target data? What do you use for $\sigma^2$? How do you choose it?
- Information about the periodic boundary condition (PBC) is completely missing. Literally the only information that the manuscript gives is *"We implement periodic boundary condition (PBC) to simulate the entire planet"*. How this is implemented is not discussed.
- Similarly, it's not clear to me how/where the "harmonic embeddings" are used in PACE. The diagram in Fig. 2 doesn't show them and only their definition is stated in the manuscript itself. What do you use for $t$? Also, the section title says "Harmonics Spatio-Temporal Embeddings" but their definition suggests that they're temporal at most.
- Fig. 2 diagram indicates that a "Adaptive Pooling" module is used at the end of PACE. I could not find any information about this module anywhere else in the manuscript.
- I presume that including such a module is important because the "spatial attention map" outputs (which in the diagram comes just before the adaptive pooling module) are squished to (0, 1) by the sigmoid function, which does not seem to match the actual range of the standardized targets.
- How exactly is a Neural ODE used inside PACE? You say that you use the dopri5 ODE solver, which is a traditional method not based on neural networks. This seems to contradict the claim that a Neural ODE is used.

A selection of typos (but note that there are many more that should be fixed):
- "medium range" -> "**medium-range**"
- "two key phenomenon" -> "two key **phenomena**"
- "modelling key physical law" -> "modelling key physical **laws**"
- Line 159: "it's" -> "**its**"
- Line 162: "descritized" -> "**discretized**"

Also, the global maps in Figures 2 and 4 are "upside-down".

References:

[1] Probabilistic Emulation of a Global Climate Model with Spherical DYffusion (https://arxiv.org/abs/2406.14798; NeurIPS 2024)

[2] The impact of internal variability on benchmarking deep learning climate emulators (https://arxiv.org/abs/2408.05288)

**Questions:**

- The advection-diffusion equation is inherently temporal, modeling atmospheric dynamics. However, the studied diagnostic-type emulation problem in this paper is not temporal in the sense that inputs and targets are from the same timestep. How do you explain this mismatch?
- Can the authors please give examples for the "few" climate emulators that "incorporate GHG concentrations typically rely on autoregressive training regimes. These models predict climate variables at future time steps based on past states, but often fail to account for the projected emissions at those future times."?
- Can you include the PACE's results, with all its components, in Fig. 5, please? It seems to me that only its ablated versions are included there.
- What are the specifics of the Neural ODE (NODE) ablation? You say that *"Neural ODE models the advection diffusion dynamics and extract features"* in the methods section, but say in the ablations section that NODE corresponds to *"remove the advection diffusion process and only parameterize the Convolution Attention Module using Neural ODE"*. These statements seem to contradict each other?
- What does Fig. 8 show? What are the reference models? Why do you show averaged TAS? Is the average over time? If so, why not just show the full time series? Why are the values negative? Is it normalized TAS that you're plotting? Which SSP is it? Can you show similar plots for more future years (not only up to 2026)?
- Can the authors expand on what they mean by *"Guan et al. (2024) proposed LUCIE (...) to account for the computational complexity of ACE."*?
- (Global) batch size of 1 is quite small. Did you accumulate gradients to alleviate the problem?
- What is the point of doing "super-emulation" if the resulting RMSEs are mostly higher than only training on the target climate model?

---

### Official Review · Reviewer_svZG · 2024-11-04

**Soundness:** 3
**Presentation:** 3
**Contribution:** 3
**Rating:** 5
**Confidence:** 4

**Summary:**

The paper introduces a lightweight 684K parameter model called Physics informed Uncertainty Aware Climate Emulator (PACE). PACE predicts the temperature and precipitation from greenhouse gases by incorporating the fundamental physics PDE of Advection Diffusion with ML models. The paper trains the PACE model on 15 climate datasets and evaluates across several climate emulators.

**Strengths:**

1. The paper is overall well-written and easy to follow.
2. Predicting uncertainty estimates is especially important for climate models, and previous SOTA methods are currently not performing UQ.
3. The proposed PACE model is significantly lightweight compared to SOTA baselines.

**Weaknesses:**

1. The motivation and description of CBAM is unclear.
2. The PACE model is only limited to surface air temperature and precipitation. Other climate models like ClimaX can predict a wide variety of target variables such as wind velocities, pressure, etc.
3. The paper does not compare against other recent SOTA climate models like FourCast [1] and GraphCast [2].

[1] Pathak, Jaideep, Shashank Subramanian, Peter Harrington, Sanjeev Raja, Ashesh Chattopadhyay, Morteza Mardani, Thorsten Kurth et al. "Fourcastnet: A global data-driven high-resolution weather model using adaptive fourier neural operators." arXiv preprint arXiv:2202.11214 (2022).

[2] Lam, Remi, Alvaro Sanchez-Gonzalez, Matthew Willson, Peter Wirnsberger, Meire Fortunato, Ferran Alet, Suman Ravuri et al. "GraphCast: Learning skillful medium-range global weather forecasting." arXiv preprint arXiv:2212.12794 (2022).

**Questions:**

1. Can the authors provide additional details on the motivation and design choices behind the CBAM layer.
2. While the paper proposes a novel method of predicting UQ using NLL, it lacks an evaluation of the uncertainty estimates. Computing simple metrics like log-likelihood and 95% confidence intervals would strengthen the work in my opinion.
3. It seems like PACE is based on the assumption that the climate is primarily driven by advection and diffusion. While it might be true on a high level, there may be more nuanced complex climate processes happening at finer scales. Can the authors comment on the limitations of modelling climate just using the advection-diffusion equation and how PACE can model these finer-grained interactions?
4. Line 240: Does Equation (15) start from $\sin(2^i. t)$ or $\sin(2^0. t)$?
5. Having an overview section after Section 3.4 to summarize the PACE architecture (in Figure 2) might improve the readability of the proposed section.
6. Table 2 should be mentioned in the paragraph starting from line 357. Further, can some more justification be given on why models like SFNO and Climax which were performing good on each individual datasets are not performing good during super emulation?

---

### Official Review · Reviewer_Ww3w · 2024-11-05

**Soundness:** 2
**Presentation:** 1
**Contribution:** 2
**Rating:** 3
**Confidence:** 4

**Summary:**

This article introduces PACE, a Physics Informed Uncertainty Aware Climate Emulator, based on Neural ODE solver focused on the advection-diffusion problem in 2D to compute the surface air temperature and precipitation.The model uses 684K parameters whereas other emulators use 107M or 200M parameters.

The authors claim to introduce Gaussian noise in order to take into account uncertainty quantification. They also claim to use periodic boundary conditions by considering Earth as a spherical domain.

The author show the results to show the generalisation capabilities on 15 climate model over 86 years.

The theoretical background is not sufficiently supported. There are many errors that lead the reader to confusion. The reader need to know more than what is written in the manuscript.

Relying on 15 emulators results, the experiments shows better accuracy in terms of RSME for surface air temperature.

The paper should be viewed as a theoretical part that requires improvements to make it clear, and experiments that highlight improvements.

**Strengths:**

The authors shows a study of the RSME regarding surface air temperature and precipitation for single emulator among . PACE outperfoms the other emulators on 13 models on 15 for temperature and 9 models over 15 for precipitation. A box plot summarize the overall performances regarding these 15 models. PACE is interesting for surface air temperature. It also use less parameters and is focused on advection-diffusion.

Ablations studies are performed with four climate model to justify the use of advection, diffusion and Neural ODE.

The authors made some claim on Green computing. This is a good step for the environment.

**Weaknesses:**

Many incorrect statements lead to confusion in the manuscript. They should be corrected carefully.

In the abstract, the author claim that numerical simulations are computationally intensive and inefficient. The second term is imprecise, the authors claims in the conclusion that their model does not capture extreme events at regional level, the training having coarse resolution. Which means numerical simulations are still essential.

In the part 2.2, the authors are talking about the work of Chen et al 2018 which should be adapted for spherical mesh. This paragraph is unclear about if the improvements are valid for spherical use, and that the authors wants to adapt to their context.

The equation 1 is unclear, v and D depend on time and space.

In the equation 3, v and D depend on x,y and t. This should be clarified.

Uncertainty quantification is claimed in the title of the paper. In the results, it is showed on the figure 3 with box plot for a whole simulation and 15 models, and in the figure 4 to show the error of approximation. The uncertainty is not quantified in the remaining figures, especially in figure 1 and 8 where Gaussian noise is used.

The Gaussian noise hypothesis should be more justified. Does this comes from a Central Limit Theorem? What is uncertainties are the authors talking about?

Line 217, 219 and 254 three \sigma are used, I assume these are three different quantities and function.

In figure 4, the discretisation difference and the error are important. The authors claim that they want to improve this later.

In section 3.2.1, the scheme for the concentration C should be clarified. This is very difficult to understand.

As stated in the section 3, Earth is presented as a rectangle in section 3. and then Spherical Fourier Neural Operator and L(i) are introduced in section 4.. This is crucial and should be improved so the reader understand that the spherical geometry is well considered even with the use of section 3.

The figure 2 needs to be improved. I would use more math symbols to be consistent with the manuscript.

References regarding the models which PACE is compared against would help to understand briefly what they are.

Regarding the results of the super emulator, no reference is done to the table 2. The reader has to look for the table.

Here are some limitations : the training of PACE use simulated data, and not real observations. Authors want to extend their study to higher resolution and using physical constraint (energy, mass and water conservation in the loss function).

The authors claims regarding Green computed are too general. Quantifying and comparing the carbon footprint and energy consumption will be appreciated.

Minor comments that do not impact the score :

Why did chose TAS for surface air temperature, instead of SAT?

Line 100, Fourier is missing in Spherical Neural Operator

At line 134, the symbol \in should be removed.

Line 170, $t$ is necessary

Line 183, co-efficient should be corrected

Line 196-197, do v_x and v_y depend on x,y and t?

Line 309, it is common to use either \sum_i or \sum_{i= 1}^N

Line 316, $lat(i)$ and $i$-th should be used

Line 440, FVM and FEM are not explained

**Questions:**

Why did the authors chose to a 86 years forecasting?

In the table 1, in the 'training resources' column hours are mentioned whereas not for the others emulator, why?

Line 54, what do the authors mean by 'compute-efficient'?

Line 101, what do the author mean by 'computational complexity'?

At line 123-124, what does continuous sequence to sequence means?

At line 133, the notation R^{x \times y} in inappropriate since x and y. Is F(t) in 2D? does the notation \mathbf{F} refers to a vector?

At line 137, is \theta a vector or a scalar?

Could you give a reference to justify the advection-diffusion process in part 3.2?

In the equation 6 and 7, what is f_\theta? Should it be introduced before?

In the part 3.2.1, where does empirical estimation of the diffusion coefficient comes form?

In equation 8, do the authors implicitly assume the concentrations C_i are independent?

Where and how the Negative Log-Likelihood is used?

As stated in the line 233, the periodic boundary conditions are imposed on latitude and longitude. This mean that the authors are identifying the North and South pole. Could you clarify that?

Could you clearly justify the use of equation 13 that account for uncertainty in the climate model?

What does the authors means by 'global spatial dependencies' at line 246? Why are AvgPoll and MaxPool used? Can you give a reference or explain more?

In the figure 2, where do the author consider the uncertainty?

What does SSP means, at line 299?

In section 4.3., are the L(i) part of SFNO?

In section 4.5, what does 'training a single model on all of 15 climate models' means? Is it what is stated after this sentence

Why the authors selected the 15 emulators AWI-CM-1-1-MR, ... , TaiEMS1 for the benchmark rather than other?

Figure 6 and 7, the title is not finished 'namely :'. Why did the authors chose bars instead of table of numbers? Especially for precipitation, the results are difficult to quantify precisely.

---

### Note · Authors · 2024-11-28

**Comment:**

We would like to withdraw our submission. We thank the reviewers for their time and effort.

**Withdrawal Confirmation:**

I have read and agree with the venue's withdrawal policy on behalf of myself and my co-authors.